# Comprehensive Metabolomic Comparison of Five Cereal Vinegars Using Non-Targeted and Chemical Isotope Labeling LC-MS Analysis

**DOI:** 10.3390/metabo12050427

**Published:** 2022-05-10

**Authors:** Zhihua Li, Chi Zhao, Ling Dong, Yu Huan, Miwa Yoshimoto, Yongqing Zhu, Ipputa Tada, Xiaohang Wang, Shuang Zhao, Fengju Zhang, Liang Li, Masanori Arita

**Affiliations:** 1Institute of Agro-Products Processing Science and Technology, Sichuan Academy of Agricultural Sciences, Chengdu 610066, China; zhaochizc@163.com (C.Z.); sophia.dl@hotmail.com (L.D.); Zhuyongqing@126.com (Y.Z.); zhangfengju27@163.com (F.Z.); 2China Application Service Center, SCIEX Analytical Instrument Trading Co., Shanghai 200335, China; yu.huan@sciex.com; 3Bioinformation and DDBJ Center, National Institute of Genetics, Mishima 411-8540, Japan; yoshimotom@nig.ac.jp (M.Y.); tipputa@reifycs.com (I.T.); 4Department of Chemistry, University of Alberta, Edmonton, AB T6G 2G2, Canada; xi14@ualberta.ca (X.W.); szhao1@ualberta.ca (S.Z.); 5RIKEN Center for Sustainable Resource Science, Yokohama 230-0045, Japan

**Keywords:** cereal vinegar, GC-MS, UHPLC-QTOF-MS, chemical isotope labeling, metabolomics, small peptides

## Abstract

Vinegar is used as an acidic condiment and preservative worldwide. In Asia, various black vinegars are made from different combinations of grains, such as Sichuan bran vinegar (SBV), Shanxi aged vinegar (SAV), Zhenjiang aromatic vinegar (ZAV), and Fujian Monascus vinegar (FMV) in China and Ehime black vinegar in Japan (JBV). Understanding the chemical compositions of different vinegars can provide information about nutritional values and the quality of the taste. This study investigated the vinegar metabolome using a combination of GC-MS, conventional LC-MS, and chemical isotope labeling LC-MS. Different types of vinegar contained different metabolites and concentrations. Amino acids and organic acids were found to be the main components. Tetrahydroharman-3-carboxylic acid and harmalan were identified first in vinegar. Various diketopiperazines and linear dipeptides contributing to different taste effects were also detected first in vinegar. Dipeptides, 3-phenyllactic acid, and tyrosine were found to be potential metabolic markers for differentiating vinegars. The differently expressed pathway between Chinese and Japanese vinegar was tryptophan metabolism, while the main difference within Chinese vinegars was aminoacyl-tRNA biosynthesis metabolism. These results not only give insights into the metabolites in famous types of cereal vinegar but also provide valuable knowledge for making vinegar with desirable health characteristics.

## 1. Introduction

As a fermented acidic condiment and preservative, vinegar has been used worldwide for more than 5000 years [1]. White vinegar is produced mainly from diluted alcohol [2], whereas black vinegar is produced from whole grains to elicit their distinct characteristics. In China, four famous black vinegar brands exist: Sichuan bran vinegar (SBV: 保寧醋), Zhenjiang aromatic vinegar (ZAV: 鎮江香醋), Shanxi aged vinegar (SAV: 山西老陳醋), and Fujian Monascus vinegar (FMV: 永春老醋). In the production of Chinese vinegar, prepared grain starters are first saccharified and then fermented to produce alcohol and acetic acid simultaneously [3]. Although the basic chemistry is the same, starters and ingredients differ substantially among the four brands.

The starter of SBV is bran koji (Fu Qu in Chinese) with wheat, rice, glutinous rice, and more than 60 medicinal herbs. After the starter fermentation is matured, uncooked wheat bran and corn are added and aerobically mixed for more than one month. After stirring to produce acetic acid, the fermented product is stored in a closed jar for maturation for at least six months before leaching. The starter of ZAV is koji (Da Qu in Chinese) with wheat, barley, and pea. The starter is mixed with cooked glutinous rice and is fermented in a fed-batch culturing style with wheat bran and rice hull in an open concrete basin with mechanical mixing. The maturation period is at least three months. The starter of SAV is similar to that of ZAV, with barley and pea at a ratio of 7:3. The starter, amounting to 60% of the raw materials, is mixed with sorghum for fermentation. Half of the material fermentation matured above is heated in a closed jar at 80~90 °C for 5~6 days (fuming) and then mixed with the remaining half for leaching. The filtrate is then matured for at least six months. Lastly, the starter of FMV is red koji (Hong Qu in Chinese) with red yeast japonica rice. The starter is mixed with steamed glutinous rice, and fermentation takes three years using fed-batch culturing in 1-year intervals.

In Japan, a different type of black vinegar (JBV) is produced [4,5,6]. It uses steamed rice with koji and yeast to produce alcohol first. Then, the broth is filtered before acetic acid fermentation takes place in a steel tank for 3 months. There is no fed-batch culturing or simultaneous fermentation of alcohol and acetic acid.

Non-targeted metabolic approaches, typically based on GC-MS and LC-MS, have been applied widely in various food analyses, such as tea [7] and milk [8], providing a view of metabolic composition. Chemical isotope labeling (CIL) LC-MS is a high-performance and quantitative profiling method. It is based on chemical derivatization using a “divide-and-conquer” strategy in which different labeling methods are used to analyze different chemical groups, including amine/phenol, hydroxyl, carboxyl, and carbonyl metabolome groups [9].

In this study, we used a combination of GC-MS, conventional LC-MS, and CIL LC-MS for a comprehensive metabolome analysis of the five famous black vinegars in China and Japan. We aimed to identify and reveal broader chemical classes, compare the different metabolites, and reveal responsible differentiating metabolic pathways in cereal vinegar. These results not only clarify metabolites in cereal vinegar but also provide valuable knowledge for improving the quality of vinegar.

## 2. Results

### 2.1. General Information of Identified Metabolites

A total of 1285 metabolites were identified with high confidence (Figure 1), in which 1190, 97, and 68 metabolites were identified by CIL LC-MS, label-free LC-MS, and GC-MS methods, respectively. Among them, 1130 (accounting for 87.9%), 67 (5.2%), and 28 (2.3%) metabolites were uniquely identified by CIL LC-MS, label-free LC-MS, and GC-MS, respectively. Among the common metabolites detected by two or more methods, 30 metabolites were identified by both CIL LC-MS and GC-MS, and 20 were identified by both CIL LC-MS and label-free LC-MS. Only 10 metabolites (<1% in total) were identified by all three methods. These results indicate that CIL LC-MS, label-free LC-MS, and GC-MS are complementary techniques for comprehensive metabolomic analysis.

In this study, we focused on determining the common metabolites found in all five types of vinegar and then examining their relative abundance differences inferred by ion intensities. We did not explore unique metabolites found in each type of vinegar. In a future study, it will be interesting to characterize unique metabolites that may play major roles in differentiating nutritional values and tastes.

### 2.2. Non-Targeted-Based Analysis of Metabolites

When the metabolites detected by GC-MS and label-free LC-MS were analyzed using a principal component analysis (PCA) biplot, the five vinegars were largely separated into three groups: JBV, FMV, and the rest (ZAV, SAV, and SBV) (Figure 2). The PCA scree plots are shown in Appendix A. The first component (PC1) explained ca. 40% of the variability and separated JBV and FMV from the rest. The second component (PC2) explained > 20% of the variability and separated JBV from FMV. In the GC-MS data, the top 10 contributing metabolites were plant-oriented saccharides and branched-chain amino acids: lactic acid (9), DL-valine (14), L-isoleucine (30), 2-hydroxyhexanoic acid (35), L-threonic acid (108), 3-phenyllactic acid (118), xylitol (145), D-xylopyranose (151), β-arabinopyranose (165), and L-iditol (209) (Figure 2I). In the label-free LC-MS data, the top 10 metabolites were amino acids and dipeptides, especially leucine in positive mode and phenolics in negative mode (Figure 2II,III). Furthermore, the PCA biplot showed that JBV, FMV, and SBV could be separated only by these metabolites in the GC-MS data (Figure 2I). Meanwhile, JBV and FMV could be separated by label-free LC-MS in positive mode (Figure 2II). 3-Phenyllactic acid (Phla) was identified by both GC-MS and label-free LC-MS in negative mode, where the ion abundances of ZAV, SAV, and SBV were much higher than those of FMV and JBV (Figure 2I,III).

For the GC-MS data (Table 1 and Appendix A), monosaccharides (glucose, fructose, and lyxose) were identified with high ion intensities, of which the highest intensity was glucose (4.49 × 10^7^, 4.65 × 10^7^, 0.66 × 10^7^, 3.52 × 10^7^, and 6.14 × 10^7^ in FMV, ZAV, SAV, SBV, and JBV, respectively), followed by fructose (3.76 × 10^7^, 2.71 × 10^7^, 0.14 × 10^7^, 4.58 × 10^7^, and 0.43 × 10^7^, respectively). Five sugar alcohols were detected: glycerol, iditol, mannitol, xylitol, and threitol. Glycerol was the most abundant in terms of ion intensity (1.01 × 10^7^, 2.56 × 10^7^, 1.68 × 10^7^, 2.88 × 10^7^, and 2.08 × 10^7^, respectively). Six amino acids and derivatives consisted of pyroglutamic acid, glycine, leucine, valine, alanine, and 4-aminobutyric acid (GABA). Among organic acids, lactic acid was the most abundant (3.31 × 10^7^, 6.32 × 10^7^, 5.13 × 10^7^, 7.56 × 10^7^, and 0.57 × 10^7^, respectively).

For the metabolites detected by label-free LC-MS (Table 1, Appendix A), 4-vinylphenol, phenylalanine, tyrosine, N-(1-deoxy-1-fructosyl)phenylalanine (0.30 × 10^7^, 0.10 × 10^7^, 0.01 × 10^7^, 2.65 × 10^7^, and 2.51 × 10^7^, respectively), isoleucine, cyclo(Pro-Leu), cyclo(Phe-Pro) (0.81 × 10^7^, 3.04 × 10^7^, 4.85 × 10^7^, 4.42 × 10^7^, and 0.89 × 10^7^, respectively), 9,10,13-trihydroxystearic acid (1.50 × 10^7^ in ZAV and 3.96 × 10^7^ SBV), tetrahydroharman-3-carboxylic acid, and harmalan (2.55 × 10^7^ in FMV and 1.20 × 10^7^ in SBV) were identified in positive mode. 3-Phenyllactic acid (2.70 × 10^7^, 3.40 × 10^7^, 3.36 × 10^7^, 3.29 × 10^7^, and 0.34 × 10^7^, respectively) was detected as the most abundant metabolite in negative mode. N-(1-Deoxy-1-fructosyl)phenylalanine was identified in both positive mode with adduct [M + H]^+^ and negative mode with adduct [2M − H]^−^. Diketopiperazines such as cyclo(Pro-Leu) and cyclo(Phe-Pro) were also identified. 9,10,13-Trihydroxystearic acid was annotated by SIRIUS4 as the top candidate in both positive mode with adduct [M + H]^+^ (71.01%) and negative mode with adduct [M − H]^−^ (77.94%) simultaneously (Table 1 and Appendix A).

### 2.3. CIL LC-MS Metabolites and LC-UV Quantification

A total of 1190 metabolites were identified based on the accurate mass and retention time (tier 1 level) and the accurate mass and predicted retention time (tier 2 level) (Appendix A). PCA biplot analysis showed that the five vinegars could be separated, particularly for JBV, FMV, and the rest (ZAV, SAV, and SBV) (Figure 3I), which is the same as those revealed by GC-MS and label-free LC-MS analyses. The 10 most contributing metabolites for PCA, namely, glutamyl-valine (A-613), leucyl-aspartate (A-721), 4-hydroxyphenylethanol (A-2109), tetrahydroharmol (A-2031), valyl-lysine (A-2231), tyrosine (A-2824), 2,4,6-triaminotoluene (A-3301), salsoline-1-carboxylic acid (A-3305), 2(N)-methyl-norsalsolinol (H-1360), and N-acetyl-tyramine (H-1649) (Figure 3I). Interestingly, 2(N)-methyl-norsalsolinol, 2,4,6-triaminotoluene, N-acetyl-tyramine, and tetrahydroharmol, were close to FMV, while leucyl-aspartate, salsoline-1-carboxylic acid, glutamyl-valine, and tyrosine were close to JBV. Furthermore, PLS-DA was used to differentiate JBV, FMV, and the rest (ZAV, SAV, and SBV), showing Q2 = 0.980 and R2 = 0.990. The group separation was very clear from the PLS-DA model, and therefore, we did not pursue further multivariate analysis using other models. As shown in Figure 3II, among the top 20 values of variable importance for the projections (VIP), 10, 8, and 2 metabolites had the highest contributions in FMV, JBV, and the rest group, respectively. Specifically, 3-sulfino-L-alanine (A-10), 2-(2-aminopropanamido)pentanoic acid (H-515), lactose (K-51), alanyl-alanine(A-386), cis,cis-muconic acid (C-1538), glycero-3-phosphoethanolamine (A-2), saccharopine (A-261), 1,3-diaminopropane (A-2572), benzhydrol (H-815), and prolyl-valine (A-1233) showed higher peak intensities in FMV, while diaminopimelic acid (A-1440), prolyl-alanine (A-1069), phenylalanyl-lysine(A-2524), tyrosyl-serine (A-2268), guaiacol (A-2819), glycyl-arginine (A-34), adenine (A-918), and 2-aminooctanoic acid (A-2753) had higher peak intensities in JBV. However, only two metabolites (i.e., seryl-lysine (A-1515) and sinapoyl aldehyde (A-2776)) had higher intensities in the rest group (ZAV, SAV, and SBV). These metabolites could be used as metabolic markers for comparing the differences among FMV, JBV, and the rest (ZAV, SAV, and SBV).

On the other side, from the metabolite concentration perspective, LC-UV quantification showed that the total concentration of dansyl-labeled metabolites in JBV (59.03 mM) was the highest, followed by SAV (41.21 mM), ZAV (35.24 mM), SBV (15.70 mM), and FMV (15.54 mM) (Figure 4). In this work, we did not perform the absolute quantification of individual metabolites. However, although CIL LC-MS is a relative quantification method used for comparing the concentration difference of a given metabolite among samples, the absolute peak intensity can be used to estimate the relative concentration differences among different metabolites in the same sample. This is because all CIL labeled metabolites have similar detectability or ionization efficiency, which means that the concentration of a labeled metabolite is reflected by its MS peak intensity [10]. For instance, two related polyamines, putrescine (1,4-diaminobutane) and cadaverine, had very high peak intensities (putrescine had 3.07 × 10^9^, 5.93 × 10^8^, 7.36 × 10^8^, 1.74 × 10^9^, and 4.63 × 10^7^ in FMV, ZAV, SAV, SBV, and JBV, respectively, while cadaverine had 2.22 × 10^9^, 4.79 × 10^8^, 5.14 × 10^8^, 1.31 × 10^9^, and 1.00 × 10^7^ in FMV, ZAV, SAV, SBV, and JBV, respectively), suggesting that both of them had higher concentrations in the sample than other metabolites having relatively lower signal intensities. Putrescine and cadaverine can be produced by the decarboxylation of amino acids in organisms and have been found in various dairy and other fermented foods, such as fermented vegetables, fish sauces, and cheese with high concentrations (from 549 mg/kg to 1560 mg/kg) [11]. In addition, another monoamine compound, tyramine, also had a high peak intensity (1.38 × 10^9^, 1.57 × 10^8^, 1.59 × 10^8^, 6.08 × 10^8^, and 1.91 × 10^6^ in FMV, ZAV, SAV, SBV, and JBV, respectively), and a related metabolite, tyrosine, was also detected with a high intensity (1.25 × 10^8^, on average) (Appendix A). Moreover, several amino acids had high intensities, including alanine (1.81 × 10^9^), proline (1.12 × 10^9^), leucine (9.96 × 10^8^), glycine (8.39 × 10^8^), valine (7.68 × 10^8^), threonine (2.60 × 10^8^), serine (2.30 × 10^8^), lysine (1.48 × 10^8^), and tyrosine (1.25 × 10^8^). Gamma-aminobutyric acid (6.74 × 10^8^), glutamic acid (3.49 × 10^8^), and aspartic acid (2.89 × 10^8^) were also identified with high peak intensities.

Several organic acids, including lactic acid, pyroglutamic acid, 4-aminobutyric acid (GABA), 3-phenyllactic acid, and hydroxyphenyllactic acid, were detected and identified with high ion intensities, likely having high concentrations, by GC-MS, label-free, and CIL-LC-MS analysis. Other non-volatile organic acids, including 5-aminopentanoic acid (6.71 × 10^7^, on average), 3-hydroxymandelic acid (5.87 × 10^7^), 4-hydroxybenzoic acid (4.07 × 10^7^), and vanillic acid (2.30 × 10^7^), also had relatively high intensities.

### 2.4. Pathway Annotation in Vinegar Metabolome

To reveal the main metabolic difference among FMV, JBV (Japan), and the rest (ZAV, SAV, and SBV, all from China), the metabolites identified by CIL LC-MS with high confidence (tier 1 and tier 2) were used as inputs for pathway analyses using MetaboAnalyst (https://www.metaboanalyst.ca/, accessed on 20 March 2021) [12]. As shown in Figure 5, the main difference between Chinese vinegars and Japanese vinegar (Figure 5I, JBV vs. FMV and the rest) was tryptophan metabolism, followed by the citrate cycle (TCA cycle) and lysine degradation, while the main difference between FMV and the rest was aminoacyl-tRNA biosynthesis (Figure 5II).

## 3. Discussion

Vinegar is rich in bioactive compounds and considered to exhibit physiological effects such as antioxidation, the prevention of obesity, and the regulation of blood pressure [6]. The integration of non-targeted and chemical isotope labeling (CIL) approaches allows for the detection of a large number of metabolites, shedding light on the metabolic composition of this complex food matrix.

From the non-targeted analysis using a combination of GC-MS and label-free LC-MS, a total of 155 metabolites, including amino acids and derivatives, di(oligo)peptides, hexoses, organic acid, etc., were annotated (Appendix A). As shown in Table 1, Appendix A, the metabolites with high ion intensities included tyramine and tyrosine. Tyramine is a bioactive amine produced through the decarboxylation of tyrosine by microorganisms such as lactic acid bacteria (LAB) [13]. Both tyramine (1.38 × 10^9^, 1.57 × 10^8^, 1.59 × 10^8^, 6.08 × 10^8^, and 1.91 × 10^6^ in FMV, ZAV, SAV, SBV, and JBV, respectively) and tyrosine (1.25 × 10^8^, on average) were detected with high intensities by CIL LC-MS (Appendix A). Notably, tyramine may cause consumer health issues such as headaches, hypertension, respiratory disorders, etc., when consuming more than 100 mg per person per meal [14].

Several amino acids and derivatives, including pyroglutamic acid (pGlu), glycine, leucine, valine, alanine, and 4-aminobutyric acid (GABA), were identified with high intensities by GC-MS and label-free LC-MS, which were further confirmed by CIL LC-MS, where pyroglutamic acid (2.04 × 10^7^), glycine (8.39 × 10^8^), leucine (9.96 × 10^8^), valine (7.68 × 10^8^), alanine (1.81 × 10^9^), and GABA (6.74 × 10^8^) showed high intensities (Appendix A). Moreover, other amino acids were identified with high intensities (from 1.48 × 10^8^–1.12 × 10^9^) by CIL LC-MS, including lysine, serine, threonine, aspartic acid, glutamic acid, and proline.

Cyclo(pro-leu) and cyclo(phe-pro) are cyclic (pro)-containing dipeptides formed by a dehydration–condensation reaction, and they have been detected in various fermented and heated foods, in which they were recognized as not only flavor compounds but also bioactives; for example, cyclo(pro-leu) could deter influenza infection [15]. Moreover, other cyclic (pro)-containing dipeptides such as cyclo(pro-val) (cosine value = 0.933), cyclo(pro-pro) (COSMIC = 0.953), and cyclo(his-pro) (COSMIC = 0.948) were also annotated with high confidence (Appendix A). Linear dipeptides were also identified [16]. They greatly affect the taste of vinegar but have received relatively little attention in previous research [4,6]. More than 150 dipeptides were confirmed by CIL LC-MS. Examples were seryl-serine (A-63) and leucyl-leucine (A-2440) for FMV, glutamyl-lysine (A-1624), valyl-asparagine (A-283), alanyl-lysine (A-1946), arginyl-alanine (A-44), and lysyl-valine (A-2269) for ZAV, SAV and SBV, and glycyl-serine (A-138) and asparaginyl-leucine (A-725) for JBV (Appendix A). Furthermore, FMV, JBV, and the rest (ZAV, SAV, and SBV) could be separated by glutamyl-valine (A-613), leucyl-aspartate (A-721), and valyl-lysine (A-2231) (Figure 3I,II). PLS-DA cross-validation details for component 1 revealed Q2 = 0.852 and R2 = 0.988, which indicate that the model is stable and reliable. This indicates that dipeptides could be used as metabolic markers for vinegar discrimination and classification. In terms of amino acid–related pathways, tryptophan metabolism was the best differentiating pathway between Japanese (JBV) and Chinese (FMV, ZAV, SAV, and SBV) vinegars, and aminoacyl-tRNA biosynthesis was the most active within the Chinese vinegars, as inferred from the KEGG Ontology categories (Figure 5).

Pyroglutamic acid (pGlu), a λ-lactam ring amino acid formed via cyclization of glutamine, was present in various types of vinegar. Its structure was reported to affect digestive absorption [17]. Besides pyroglutamic acid, several other organic acids, including lactic acid, 3-phenyllactic acid, hydroxyphenyllactic acid, GABA, 5-aminopentanoic acid, 3-hydroxymandelic acid, 4-hydroxybenzoic acid, and vanillic acid, were identified as the main organic acids in this study (Table 1, Appendix A). Lactic acid had the highest intensity (3.31 × 10^7^, 6.32 × 10^7^, 5.13 × 10^7^, 7.56 × 10^7^, and 0.57 × 10^7^, respectively) in GC-MS, followed by 3-phenyllactic acid (2.70 × 10^7^, 3.40 × 10^7^, 3.36 × 10^7^, 3.29 × 10^7^, and 0.34 × 10^7^, respectively) and hydroxyphenyllactic acid (1.65 × 10^7^, 0.94 × 10^7^, 0.50 × 10^7^, 0.42 × 10^7^, and 0.51 × 10^6^, respectively) in the negative mode of UHPLC-QTOF-MS, which are in accordance with the results obtained by CIL LC-MS (Appendix A). 3-Phenyllactic acid (Phla), possibly produced by LAB and yeast (data not shown), showed broad-spectrum and effective antimicrobial properties [18,19]. Peters et al. reported that D-Phla, produced by LAB, was absorbed by the human gut and is involved in the regulation of immune functions through the activation of hydroxycarboxylic acid receptor 3 (HCA3) [20]. Notably, 3-phenyllactic acid was located close to the rest group (ZAV, SAV, and SBV) by GC-MS and label-free LC-MS in negative mode, while tyrosine was close to JBV, as confirmed by both label-free LC-MS in positive mode and CIL LC-MS. These results indicate that 3-phenyllactic acid and tyrosine could also be used as metabolic markers for vinegar discrimination and classification, in addition to the dipeptide markers.

From the metabolic analysis, we could also make an inference about the manufacturing process. The detection of an Amadori product N-(1-deoxy-1-fructosyl)phenylalanine, also identified in tea [21] and botanical extracts [22], implied the heating (Maillard) process of vinegar production. Its amount was the highest in JBV and the lowest in SAV.

Tetrahydroharman-3-carboxylic acid and harmalan, belonging to aromatic β-carboline alkaloids, can be found in traditional medicinal plants and kanjang (Korean soy source), exhibiting biological activities such as antimitotic activity [23] and anti-neuroinflammatory activities [24], and they might originate from the raw materials used in making the vinegar.

## 4. Materials and Methods

### 4.1. Materials

LC-MS grade water, methanol (MeOH), acetonitrile (ACN), 0.1% formic acid (FA) in water, 0.1% FA in ACN, GC-MS derivatizing agents including methoxyamine hydrochloride, pyridine, and *N*-methyl-*N*-(trimethylsilyl) trifluoroacetamide (MSTFA) were purchased from Sigma-Aldrich (Sigma-Aldrich Shanghai Trading Co., Ltd., Shanghai, China or Markham, ON, Canada) unless indicated otherwise.

All vinegar brands were bottled commercial products available nationwide in China. Sichuan bran vinegar (SBV, 6.91% acidity, Sichuan Baoning Vinegar Co, Ltd., Langzhong, China), Shanxi aged vinegar (SAV, 7.03% acidity, Shanxi Laochen Vinegar Group Co., Ltd.), Zhenjiang aromatic vinegar (ZAV, 6.28% acidity, Jiangsu Hengshun Vinegar Industry Co., Ltd., Zhenjiang, China), and Fujian monascus vinegar (FMV, 6.99% acidity, Fujian Yongchun Old Vinegar Industry Co., Ltd., Yongchun, China) were produced in Sichuan province, Shanxi province, Jiangsu province, and Fujian province, respectively. Japanese black vinegar (JBV, 4.56% acidity, Moribun Co., Ltd., Kita-gun, Japan) was produced in Japan (Figure 6).

### 4.2. GC-MS Analysis

A 100 µL vinegar sample was extracted with 1 mL of mixed solvent (isopropanol:acetonitrile:water = 3:2:2, *v*/*v*/*v*) in a 2 mL Eppendorf tube using a Retsch MM400 Mixer Mill (Retsch GmbH&Co, Germany) at 30 Hz/s for 3 min and then centrifuged at 12,000× *g* for 5 min at 4 °C. After centrifugation, 200 µL of the supernatant was transferred to a new 1.5 mL Eppendorf tube and freeze-dried using a vacuum freeze drier (FDU-2110, EYELA, Japan). The dried sample was subjected to a two-step derivatization process with 100 µL of methoxyamine hydrochloride in pyridine (20 mg/mL) at 37 °C for 90 min, followed by 100 µL of MSTFA containing 1% trimethylchlorosilane at 37 °C for 90 min. The derivatized vinegar samples (1 µL) were injected into an Agilent GC 9000 coupled to 5977B mass-selective detector (MSD) (Agilent Technologies, St Louis, MA, USA). The separation was carried out on a DB-5MS fused silica capillary column (30 m × 0.25 mm × 0.25 μm film thickness, J&W Scientific, Folsom, CA, USA). The helium flow rate was 1.0 mL/min, and analysis was conducted in split mode (25:1). The column temperature was initially maintained at 80 °C for 2 min, raised from 80 °C to 200 °C at 4 °C/min, held at 200 °C for 1 min, then increased from 200 °C to 230 °C at 5 °C/min, and maintained at 230 °C for 5 min. The other MSD conditions were as follows: injector temperature was 250 °C, electron energy was 70 eV, and mass range was 20–600 amu. Compounds were identified using the NIST17 library as well as by comparing their retention indices (RIs) in the NIST Chemistry WebBook (http://webbook.nist.gov/, accessed on 15 January 2020). For relative intensity comparison, signal intensity of a metabolite was normalized by the total ion chromatogram (TIC) intensity. Each sample was analyzed in triplicate (*n* = 3).

### 4.3. Conventional or Label-Free LC-MS Analysis

The sample was centrifuged at 12,000× *g* for 10 min, and the supernatant was ultrafiltered through a 10 kDa cut-off filter device (Vivacon 500, Sartorius Stedim Lab Ltd, Stonehouse, UK). Samples were analyzed on a Shimazu L30A ultrahigh-pressure liquid chromatography (UHPLC) system coupled to a TripleTOF™ 5600 mass spectrometer (AB SCIEX, Foster City, CA, USA) equipped with a DuoSpray ion source. Chromatographic separation of the vinegar samples was performed on Zorbax Eclipse Plus C18 column (100 × 2.1 mm, 1.8 μm, Agilent Technology, Little Falls, DE, USA). The column was maintained at 40 °C, and the injection volume was 3 µL. Compounds were eluted using a binary mobile phase at a flow rate of 0.5 mL/min. The mobile phase contained solvent A (0.1% formic acid in water (*v*/*v*)) and B (0.1% formic acid in acetonitrile (*v*/*v*)). An elution linear gradient program was performed as follows: 0–5 min, 10% B; 5–25 min, 10–50% B; 25–33 min, 50–95% B; 33–38 min, 95% B; 38–39 min, 95–10% B; 39–45 min, 10% B. The DDA (data-dependent acquisition) parameters in both positive (ESI+) and negative (ESI−) ion modes were set as: Gas1, 50 psi; Gas2, 50 psi; curtain gas, 35 psi; source temperature, 500 °C; capillary voltage, 4.5 kV; cone voltage, 45 ± 20 V; TOF MS scan *m*/*z* range: 50–1250 Da; product ion scan *m*/*z* range: 50–1250 Da; MS1 accumulation time, 0.15 s; MS2 accumulation time, 0.06 s; cycle time, 0.8 s; and dependent product ion scan number, 10. Automatic mass calibration was performed every 5 injections using a calibration delivery system (CDS) with atmospheric pressure chemical ionization (APCI) positive/negative calibration solutions. Each sample was analyzed in four replicates (*n* = 4).

The raw DDA data (.wiff format) files were converted to ABF format using Reifycs Abf (Analysis Base File) Converter (Reifycs Inc., Tokyo, Japan) as input to MS-DIAL (v4.16) for peak detection and alignment [25]. The alignment results of signals normalized with total ion chromatogram (TIC) intensity were exported from MS-DIAL for relative quantification for principal component analysis. The peaks with representative MS/MS spectra were annotated by MS-FINDER (v3.30) [26], SIRIUS4 (v4.01) [27], and Global Natural Products Social Molecular Networking (GNPS) [28].

### 4.4. CIL LC-MS Analysis

Each sample was diluted to 2 mM after sample normalization using the total concentration of labeled metabolites measured by LC-UV. Sample labeling strictly followed the SOP in the provided kit from Nova Medical Testing Inc. (NovaMT) (Edmonton, AB, Canada) [10]. An Agilent eclipse plus reversed-phase C18 column (150 × 2.1 mm, 1.8 µm particle size) was used to separate the labeled sample, and Agilent 1290 LC linked to Bruker Impact II QTOF Mass Spectrometer was used to detect the labeled metabolites. Solvent A was water with 0.1% formic acid, whereas solvent B was acetonitrile with 0.1% formic acid. The LC gradient was 25% B from 0 to 10 min, 99% B from 10 to 13.1 min, and 25% B from 13.1 to 16 min, and it ended at 25% B. The flow rate was 400 μL/min. The column oven temperature was 40 °C, and mass range was 220–1000 *m*/*z*. The raw mass data were exported to csv file with Bruker DataAnalysis 4.4. Data analysis was performed using IsoMS Pro 1.2.5 (NovaMT Inc., Edmonton, AB, Canada) with the parameters: minimum *m*/*z*: 240; maximum *m*/*z*: 1000; saturation intensity: 20,000,000; retention time tolerance: 22 s; and mass tolerance: 10 ppm. Identified metabolites were searched against the in-house mass spectral library NovaMT Metabolite Database v2.0 based on the accurate mass and retention time with retention time tolerance of 30 s and mass tolerance of 10 ppm. For each sample, the analysis was performed in triplicate. For total metabolite concentration measurement, the amine/phenol labeled sample was centrifuged at 15,000 g for 10 min. An Agilent 1290 UPLC system with a photodiode array (PDA) detector was used for quantifying the total labeled or pooled sample concentration, as described in a previous study [29].

### 4.5. Pathway Analysis

To investigate and compare the metabolic pathways present in vinegar, the Kyoto Encyclopedia of Genes and Genomes (KEGG) was used as the backend knowledgebase. Non-targeted analysis of metabolites identified by a combination of GC-MS and UHPLC-QTOF-MS with high confidence (Appendix A) and the tier 1 and tier 2 metabolites identified by CIL LC-MS (Appendix A) were mapped to KEGG pathways to reveal the main pathway distributions among the different types of vinegar using MetaboAnalyst (https://www.metaboanalyst.ca/, accessed on 20 March 2021) [12].

### 4.6. Statistical Analysis

Statistical analyses were performed in Excel and other software. Principal component analysis (PCA) was carried out with correlation matrix by the FactoMineR packages in open-source software language R (version 3.6.3). Partial least-squares discriminant analysis (PLS-DA) was performed by MetaboAnalyst 5.0 (https://www.metaboanalyst.ca/, accessed on 20 March 2021) [12]. Venn diagrams were used to determine the commonly identified metabolites using chemical isotope labeling LC-MS, label-free LC-MS, and GC-MS and visualized by jvenn, an interactive Venn diagram viewer (http://www.bioinformatics.com.cn/static/others/jvenn/index.html, accessed on 20 March 2021 ) [30].

## 5. Conclusions

In this study, a combination of GC-MS, label-free LC-MS, and chemical isotope labeling LC-MS was used to profile the chemical compositions of five black vinegars. We aimed to identify and reveal broader chemical classes, compare the different metabolites, and reveal responsible differentiating metabolic pathways in cereal vinegar. Different types of vinegar showed different compositions of metabolites. Various diketopiperazines and linear dipeptides contributing to different taste effects were identified. Dipeptides, 3-phenyllactic acid, and tyrosine could be used as metabolic markers for differentiating cereal vinegars. As far as we know, cyclic peptides are reported here for the first time as vinegar metabolites and are considered important components. Hydroxyphenyllactic acid, 3-phenyllactic acid, lactic acid, and pyroglutamic acid were abundant organic acids. Amino acid–related pathways such as tryptophan metabolism and aminoacyl-tRNA biosynthesis were the top differentiating pathways. These results not only give insights into metabolic compositions in different cereal vinegars but also provide valuable knowledge for consumers and for making vinegar with desirable health characteristics.

## Figures and Tables

**Figure 1 metabolites-12-00427-f001:**
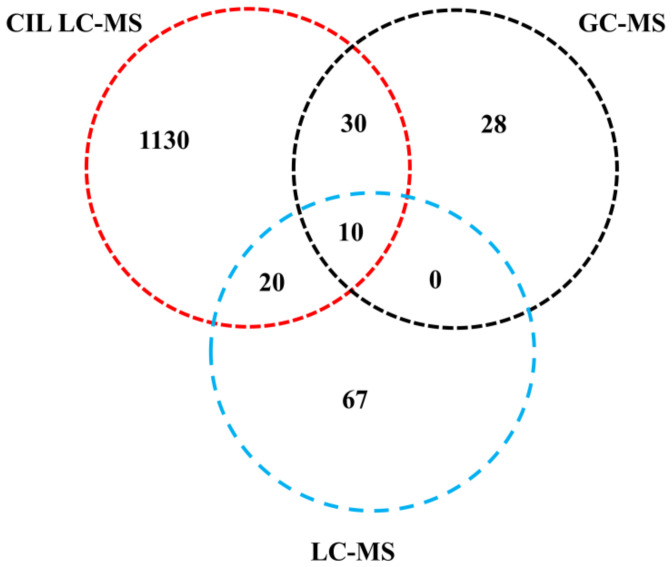
General information on identified metabolites using CIL LC-MS, label-free LC-MS, and GC-MS, respectively.

**Figure 2 metabolites-12-00427-f002:**
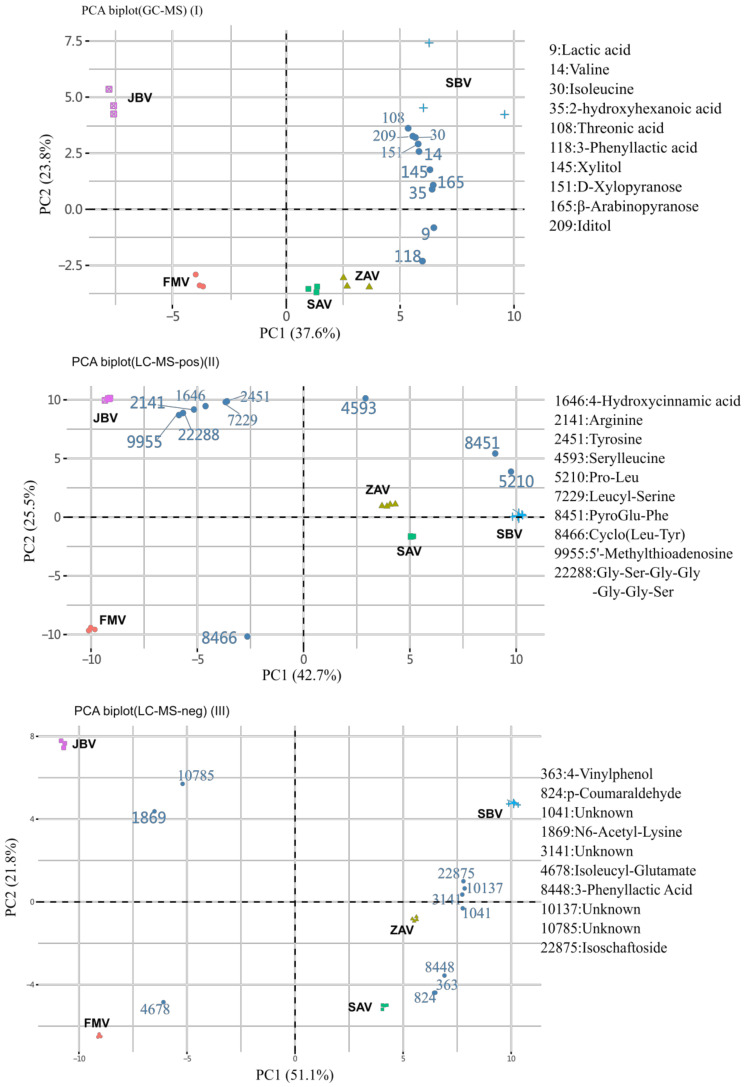
PCA biplots of vinegar metabolites based on non-targeted metabolic analysis.

**Figure 3 metabolites-12-00427-f003:**
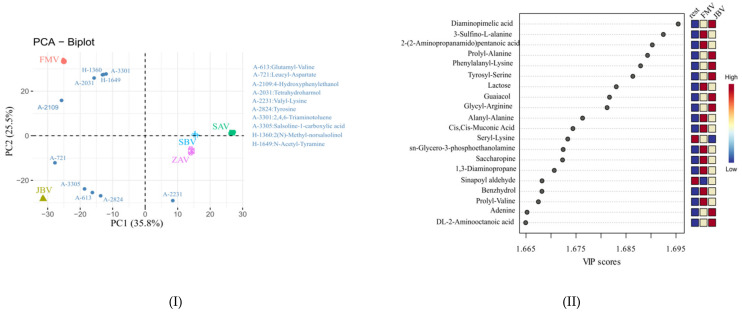
(**I**) PCA biplot of vinegar metabolites based on chemical isotope labeling LC-MS quantitative metabolomic analysis and (**II**) VIP score plot from partial least-squares discriminant analysis (PLS-DA) for FMV, JBV, and the rest (ZAV, SAV, and SBV).

**Figure 4 metabolites-12-00427-f004:**
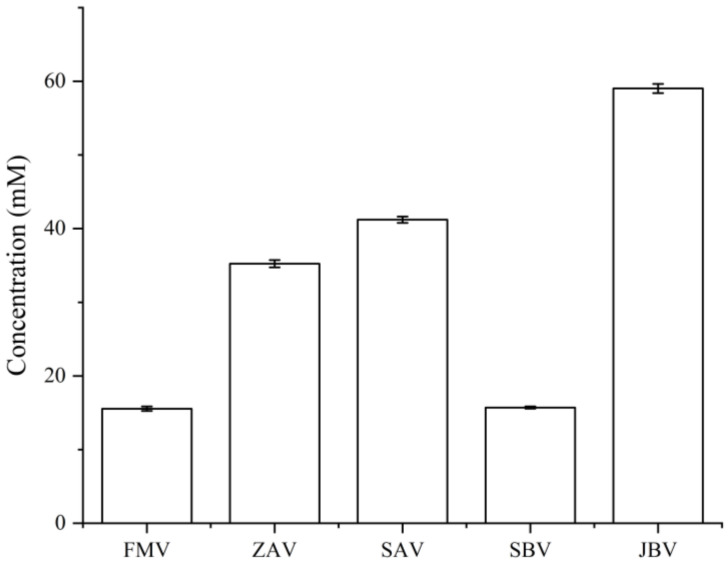
Total concentration of labeled metabolites measured by LC-UV in different vinegars. The concentration data with an error bar are presented as the mean ± SD of three replicates (*n* = 3).

**Figure 5 metabolites-12-00427-f005:**
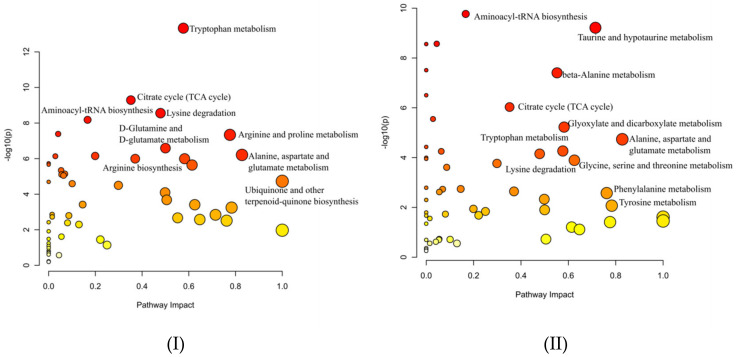
Comparison of main pathways present in the cereal vinegar.

**Figure 6 metabolites-12-00427-f006:**
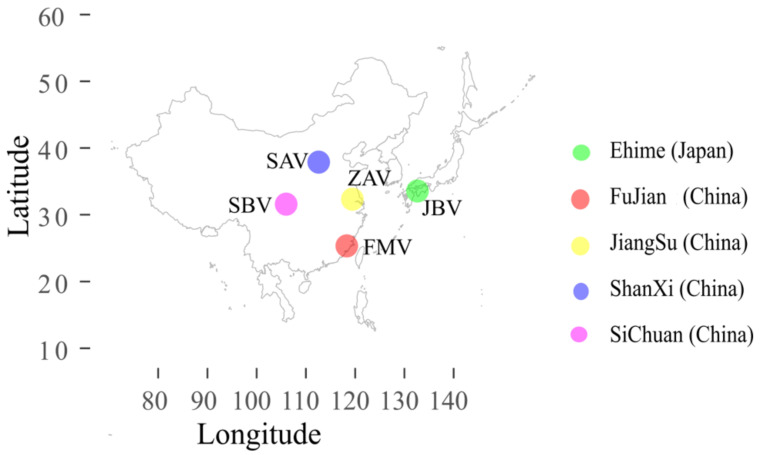
Origin of vinegar production in this study.

**Table 1 metabolites-12-00427-t001:** High-intensity metabolites identified by GC-MS and label-free UHPLC-QTOF-MS ^a^.

Metabolite	PubChem	Relative Ion Intensity in Different Types of Vinegar (×10^6^)					Method ^b^
		FMV	ZAV	SAV	SBV	JBV	
Benzene and Substituted Derivatives (2) ^c^							
Tyramine	5610	3.72 ± 0.17	0.71 ± 0.05	0.7 ± 0.07	0.73 ± 0.42	0.14 ± 0.02	G, LP
4-Vinylphenol	62,453	1.27 ± 0.01	1.83 ± 0.08	1.84 ± 0.03	1.72 ± 0.05	0.16 ± 0	LP, LN
Carboxylic acids and derivatives (14)							
Pyroglutamic acid	7405	4.81 ± 0.23	18.04 ± 1.56	15.01 ± 0.26	18.12 ± 1.36	4.29 ± 0.56	G, LN, LP
Glycine	750	6.2 ± 0.28	7.93 ± 0.4	4.92 ± 0.32	3.35 ± 1.23	2.82 ± 0.19	G
Leucine	6106	5.47 ± 0.11	7.86 ± 1.42	4.48 ± 0.13	6.55 ± 0.99	2.4 ± 0.12	G
Succinic acid	1110	4.59 ± 0.22	7.88 ± 0.27	3.76 ± 0.22	5.75 ± 1.29	2.05 ± 0.11	G, LN
Valine	6287	0.2 ± 0.06	2.66 ± 0.44	1.17 ± 0.21	5.52 ± 0.82	0.55 ± 0.06	G
Alanine	5950	8.99 ± 0.54	7.28 ± 1.2	6.88 ± 0.44	8.61 ± 1.25	1.12 ± 0.16	G
Citric acid	311	0.12 ± 0.02	2.06 ± 0.25	3.48 ± 0.01	0.14 ± 0.01	0.67 ± 0.05	G, LN
4-Aminobutyric acid(GABA)	119	3.88 ± 0.32	2.44 ± 0.21	4.64 ± 0.25	1.07 ± 0.63	0.39 ± 0.02	G
Phenylalanine	6140	0.29 ± 0.01	0.38 ± 0.01	0.27 ± 0.01	0.4 ± 0.01	0.52 ± 0.01	LP, G, LN
Tyrosine	6057	0.87 ± 0.01	2.76 ± 0.1	2.41 ± 0.12	3.46 ± 0.12	0.61 ± 0.02	LP, G
*N*-(1-Deoxy-1-fructosyl)phenylalanine	101,039,148	2.99 ± 0.18	10.35 ± 0.27	0.95 ± 0.06	26.55 ± 1.19	25.14 ± 0.92	LP, LN
Isoleucine	6306	1.1 ± 0.04	1.74 ± 0.06	1.34 ± 0.06	1.42 ± 0.07	1.53 ± 0.04	LP, LN, G
Cyclo(Pro-Leu)	102,892	0.65 ± 0.03	1.15 ± 0.02	7.23 ± 0.19	1.06 ± 0.05	0.11 ± 0.01	LP
Cyclo(Phe-Pro)	99,895	8.08 ± 0.46	30.36 ± 0.75	48.52 ± 1.12	44.23 ± 2.11	8.9 ± 0.29	LP
Hydroxy acids and derivatives (1)							
Lactic acid	107,689	33.05 ± 0.89	63.18 ± 0.61	51.33 ± 1.41	75.58 ± 0.5	5.68 ± 0.23	G, LN
Phenylpropanoic acids (2)							
3-Phenyllactic acid	3848	27.02 ± 1.05	33.97 ± 1.56	33.58 ± 1.39	32.9 ± 0.62	3.35 ± 0.03	LN, G
Hydroxyphenyllactic acid	9378	16.49 ± 0.88	9.41 ± 0.14	4.99 ± 0.11	4.19 ± 0.1	0.51 ± 0	LN
Fatty Acyls (2)							
isohexonic acid	12,344	0.68 ± 0.06	0.85 ± 0.07	0.3 ± 0.04	0.56 ± 0.05	4.5 ± 0.74	G
9,10,13-Trihydroxystearic acid	45,359,277	1.65 ± 0.05	14.97 ± 0.24	4.39 ± 2.57	39.55 ± 0.93	0 ± 0	LP, LN
Harmala alkaloids (2)							
Tetrahydroharman-3-carboxylic acid	73,530	1.03 ± 0.07	4.29 ± 0.08	2.32 ± 0.03	0.83 ± 0.05	4.55 ± 0.23	LP, LN
Harmalan	160,510	25.51 ± 0.53	0.19 ± 0.01	3.32 ± 0.15	11.97 ± 0.31	0.21 ± 0.02	LP
Organooxygen compounds (14)							
Glucose	5793	44.94 ± 2.4	46.47 ± 0.97	6.57 ± 0.83	35.19 ± 23.77	61.87 ± 1.51	G
Glycerol	753	10.08 ± 0.35	25.73 ± 1.13	16.79 ± 0.61	28.82 ± 2.64	20.75 ± 0.54	G
2,3-Butanediol	262	4.63 ± 0.19	8 ± 0.16	6.73 ± 0.19	5.91 ± 0.78	4.74 ± 0.15	G
Fructose	439,163	37.55 ± 3.58	27.07 ± 3.12	1.35 ± 0.08	45.83 ± 3.17	4.34 ± 0.4	G
Inositol	892	0.96 ± 0.22	16.33 ± 1.59	13.32 ± 1.77	11.16 ± 1.43	3.46 ± 0.31	G
Ribofuranose	5779	0.17 ± 0.02	11.72 ± 0.62	9.49 ± 1.43	5.52 ± 0.92	2.06 ± 0.15	G
Iditol	5,460,044	0.46 ± 0.4	5.38 ± 0.65	1.19 ± 0.07	10.67 ± 1.01	0.78 ± 0.3	G
Xylopyranose	135,191	0.03 ± 0	6.91 ± 0.78	10.98 ± 1.06	24.04 ± 2.77	0.58 ± 0.03	G
Mannitol	6251	0.47 ± 0.25	13.24 ± 1.38	4.71 ± 0.48	2.98 ± 0.29	0.39 ± 0.05	G
Xylitol	6912	0.07 ± 0.01	4.72 ± 0.12	4.27 ± 0.38	9.13 ± 0.86	0.24 ± 0.03	G
Threitol	222,285	2.66 ± 0.12	1.9 ± 0.16	5.23 ± 0.56	10.68 ± 0.9	0.14 ± 0.02	G
Glyceric acid	439,194	0.28 ± 0.01	2.14 ± 0.14	4.11 ± 0.28	1.65 ± 0.33	0.13 ± 0.01	G
Lyxose	439,240	0.01 ± 0	1.71 ± 0.31	2.28 ± 0.17	6.17 ± 0.89	0.09 ± 0.01	G
1,3-Propanediol	347,971	6.64 ± 0.27	0.11 ± 0.02	0.69 ± 0.01	0.91 ± 0.11	0.02 ± 0	G

^a^ The relative ion intensity with >1% average abundance in at least one type of vinegar. The metabolites tentatively identified are listed in Appendix A with more details, including retention time, mz, adducts, formula, and InChIKey. ^b^ G, LP, and LN denote GC-MS and UHPLC-QTOF-MS in positive and negative modes, respectively. If one metabolite was identified by more than one method, the higher relative ion intensity is listed with the corresponding method used to identify the metabolite. ^c^ Based on ClassyFire (http://classyfire.wishartlab.com, accessed on 15 January 2020). Numbers in each class are given in parentheses.

## Data Availability

The raw data set included GC-MS and LC-MS have been deposited in the MassIVE database (ftp://massive.ucsd.edu/MSV000089449/, accessed on 20 April 2022).

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
