# Peer review of "Comprehensive Metabolomic Comparison of Five Cereal Vinegars Using Non-Targeted and Chemical Isotope Labeling LC-MS Analysis"

_metabolites, 2022, doi:10.3390/metabo12050427_

Round 1

Reviewer 1 Report

I have no major comments, good work. 

Author Response

Thank you very much for the compliment of this work.

Reviewer 2 Report

The manuscript described a metabolomics analysis of different types of vinegar using a combination of GC-MS, conventional LC-MS and chemical isotope labeling LC-MS. A total of 1,285 metabolites were detected and various diketopiperazines and linear dipeptides were considered contributing to different taste effects in vinegar.

  1. Figure 1 described the 1285 metabolites technical belongings in a Venn diagram. Which type of vinegar containing majority of the metabolites? If possible, could the author consider the metabolites origin belonging and draw the Veen diagram as well?
  2. For the PCA analysis, what are the scree plots? Could the author attach the plots in the supplementary information?
  3. What are the Q2 and R2 results of PLS-DA model? Have the authors considered the following distinguish model to validate the model, e.g., random forest analysis?
  4. The author mentioned the relative intensity comparison in Table 1, what is the data normalization method besides TIC? Has the author considered an internal and/or external standard for GC-MS and LC-MS?
  5. For Figure 4, what is the bar for the plot, SD or SEM? If possible, could the author consider a boxplot?

Reviewer 3 Report

The Li et al. manuscript entitled “Comprehensive metabolomic comparison of five cereal vinegars using non-targeted and chemical isotope labeling LC-MS analysis” deals with the study on metabolomic (chemical) profile of five black vinegars originating from China (four products) and Japan (one product). Detailed experimental study carried out using supplementary techniques (GC-MS, LC-MS and CIL-LC-MS) and supported by advanced multivariate statistical methods (PCA and PLS-DA) identified a total number of 1285 metabolites. Two of them were identified for the first time (tetrahydroharman-3-carboxylic acid and harmalan), whereas dipeptides, 3-phenyllactic acid and tyrosine were identified as potential metabolic biomarkers differing vinegars.

Regardless of scientific value, this research also provides valuable knowledge for vinegars consumers. However, data are missing on tentative concentrations of the main metabolites in vinegars. Figure 4 shows only a total labelled metabolite concentration (mM). According to the authors’ statement “concentration of a labeled metabolite could be reflected by its MS peak intensity”. This is too vague statement. Lines 189-190 – putrescine has very high MS peak intensity (3.07 x 109) – hence, putrescine can be found on the nanoM level of concentration in vinegars?

PCA calculations were carried out on covariance or correlation matrix?

References 1, 11, 12, 17, 20, 24, 25, 26, 27, and 30 – the names of some of the authors of the publication have been omitted – it should be corrected.

It would be better if the quality of the English language was checked by a native speaker (for example, line 340 “was performed on carried out on”, line 378 “the amine & phenol”).

Author Response

Dear Reviewer,

  Please see the attached file! Thank you very much!

Best regards

Li Zhihua

Reviewer 4 Report

This manuscript is very interesting and well edited. It fits to the Metabolites journal and I think that it should be admitted to publish in this journal. However, earlier, this text must be improved and complemented by missing information. My comments are as follows:

  • Section „2. Results” – Please discuss the size of the PC1 and PC2 value received by principal component analysis (PCA) biplot (see figures 2 and 3). These values are not always high, and this is important for the interpretation of the results.
  • Section „4. Materials and Methods” – Please add the information in how many repetitions, individual analyzes were performer.
  • Section „5. Conclusions” – Please include in this section the aim of the research previously described. In this section you must answer the formulated target aim of the research.

Author Response

To Reviewer 4:

“This manuscript is very interesting and well edited. It fits to the Metabolites journal and I think that it should be admitted to publish in this journal. However, earlier, this text must be improved and complemented by missing information. My comments are as follows:

Section „2. Results” – Please discuss the size of the PC1 and PC2 value received by principal component analysis (PCA) biplot (see figures 2 and 3). These values are not always high, and this is important for the interpretation of the results.”

Response: This is because we used the call: PCA(X, scale.unit = T, graph = FALSE) in the FactoMineR packages in open-source software language R (version 3.6.3). “scale.unit = True” means data are scaled to unit variance. Although these values are not always high, they can still be used to evaluate the contribution of PCA.

“Section „4. Materials and Methods” – Please add the information in how many repetitions, individual analyzes were performer.”

Response: GC-MS and CIL-LC-MS used three replicates, while UPLC-QTOF-MS used four replicates. The information is shown on Page 12, line 359; Page 12, line 380; Page 13, line 405.

“Section „5. Conclusions” – Please include in this section the aim of the research previously described. In this section you must answer the formulated target aim of the research.”

Response: Thank you for the suggestion. We have edited the conclusions (see Page 13, lines 430-443).
